# c-D-index at day 11 can predict febrile neutropenia during chemotherapy in acute myeloid leukemia

Hiroyuki Kubo⊚, Osamu Imataki⊚ *⊚, Yukiko Hamasaki Kubo, Makiko Uemura

Division of Hematology, Faculty of Medicine, Kagawa University, Kagawa, Japan

⊚ These authors contributed equally to this work.
* imataki.osamu@kagawa-u.ac.jp

**Data Availability Statement:** All relevant data are within the manuscript and its Supporting Information files.

**Funding:** This work was supported by JSPS KAKENHI Grant Numbers JP19K17927.

## Abstract

Acute myeloid leukemia (AML) often requires long-term intensive chemotherapy for its cure. During chemotherapy, the patient always experiences neutropenia with readings below 500 cells/μL; this is often accompanied by pyrexia with a temperature of more than 101˚F. This combination of neutropenia and fever is called febrile neutropenia (FN). A tool to sum up the daily severity of a patient's neutropenia, the "D-index," has been validated in some specific clinical settings. In this study, we examined whether the D-index is a useful predictor of the onset of FN. We recruited consecutive patients treated with induction and consolidation chemotherapy for newly diagnosed AML. We gathered all the FN events and their clinical background data retrospectively. Patients' background, such as pre-existing conditions and disease status before the treatment, were analyzed using multivariate methods. All FN events during chemotherapy were evaluated for infection focus and causative organism. A total cohort of 51 cases (25 women, 26 men; median age 51 years, range 18–74) was analyzed. They displayed 171 neutropenic events (115 FN and 56 afebrile episodes) during chemotherapy, and complete neutropenic events were used in this study. Sensitivity and specificity analysis showed that the most useful cutoff value to predict the onset of FN was a cumulative D-index at day 11 (c-D11-index) of 718. The cumulative incidence of FN during chemotherapy was significantly higher in the group with c-D11-index ≥710 (80%) than in the group with c-D11-index <710 (39%) (P < 0.0001). Through multivariate analysis, the presence of diabetes mellitus and the c-D11-index were extracted as contributing factors to the onset of FN (P = 0.0087 and 0.0002, respectively). In conclusion, we can predict that AML patients receiving chemotherapy will experience the complication of FN when the c-D-index at day 11 is >710, with an odds ratio of 2.1.

## Background

Acute myeloid leukemia (AML) has been identified as a leading hematological malignancy, with an onset prevalence of 3.7 per 100,000 persons per year [1]. Among the various types of cancers, acute leukemia—including AML—requires the most intensive chemotherapy. A

**Competing interests:** The authors have declared that no competing interests exist.

chemotherapy protocol for AML consists of induction and consolidation chemotherapy [2]. These intensified chemotherapy regimens guide a substantial proportion of the population to cure status through disease remission. Unfortunately, severe neutropenia will occur during such intensive chemotherapy, with levels five times less than the normal range of neutrocytes (≤500 cells/μL), which is grade 4 neutropenia as defined by the National Cancer Institute (NCI)-Common Terminology Criteria for Adverse Events (CTCAE) v5.0.

Thus, neutropenia followed by intensified chemotherapy has been identified to be a leading risk factor for opportunistic infection [3]. Concomitance of severe neutropenia (≤500 cells/μL) with high fever (≥101°F) is called febrile neutropenia (FN) by the definition of the Infectious Diseases Society of America (IDSA) [4]. FN might cause severe infection, particularly in patients receiving chemotherapy for hematologic malignancies [5]. In half of the FN episodes, the infectious pathogens remain unclear, even though adequate and routine workup for infection is applied [5, 6]. Portugal et al. originally reported the D-index as a quantitative evaluation of the severity of a patient's neutropenia [7]. The utility of the D-indices has been investigated in some specific clinical situations [7–11]. Recent clinical trials were specifically performed for fungal infections [12–16], and the D-indices were found to be useful in guiding clinical decisions regarding antifungal therapy [14, 16].

A great deal of evidence has accumulated on the relevance of the D-index for predicting invasive fungal infections (IFIs) in patients with hematologic malignancies. However, these patients are predisposed to FN; hence, an index that predicts only pulmonary infection [9, 10] is deemed not ideal. Previously, we have reported positive relation between the c-D-index and prolonged neutropenia in an adult population [17]. While the D-index did not predict the onset and severity of FN, the c-D-index was associated with prolonged FN. In the previous study, we investigated whether c-D-index can predict FN during chemotherapy. It, however, could not achieve enough power to detect a significant correlation for predicting FN using the c-D-index. Thus, in this present study, we extended the number of adult patients included in the statistical analysis.

## Subjects and methods

### Patient selection

We recruited consecutive all AML patients treated in our hospital between November 1998 and December 2018. The inclusion criteria were as follows: (i) aged from 18 to 74 years old, (ii) Eastern Cooperative Oncology Group performance status (PS) 0–2, and (iii) hematologic diagnosis of *de novo* AML. The exclusion criteria were as follows: (i) poor PS (≥3) at the time of diagnosis, (ii) acute promyelocytic leukemia (i.e., French–American–British [FAB] classification M3), and (iii) poor general condition where the ability to tolerate intensive chemotherapy was in question.

All patients were basically treated with Japan Adult Leukemia Study Group (JALSG) 97 [18] or JALSG 201 [19] protocol regimens, as the standard therapy used in our country in practice. We have used JALSG disease risk as the risk factor at the onset of AML [18].

### Treatment regimens

The induction chemotherapy regimen consisted of 12mg/m$^2$ idarubicin daily for 3 days or 50mg/m$^2$ daunorubicin (DNR) daily for 5 days, combined with 100mg/m$^2$ Ara-C daily for 7 days. The consolidation chemotherapy regimen consisted of a total of three cycles of high-dose Ara-C at 2g/m$^2$ twice a day for 5 days. Alternatively, we used four courses of consolidation therapy as follows:

- First consolidation: Ara-C 200mg/m$^2$ daily for 5 days and mitoxantrone 7mg/m$^2$ for 3 days.

- Second consolidation: Ara-C 200mg/m$^2$ daily for 5 days and DNR 50mg/m$^2$ daily for 3 days.

- Third consolidation: Ara-C 200mg/m$^2$ daily for 5 days and aclarubicin 20mg/m$^2$ daily for 5 days.

- Final consolidation: Ara-C 200mg/m$^2$ daily for 5 days, etoposide 100mg/m$^2$ daily for 5 days, vincristine 0.8mg/m$^2$ on day 8, and vindesine 2mg/m$^2$ on day 10.

## Definition of FN and febrile episode

Body temperature was measured. FN was defined as fever $\geq$100˚F (38.0˚C) at the axilla with neutrophil count <500 cells/μL. A temperature of 100˚F (38.0˚C) at the axilla corresponds with a central temperature of 101˚F (38.5˚C) measured in the oral cavity. If a fever $\geq$100˚F (38.0˚C) was observed again after the body temperature had decreased to <99.5˚F (37.5˚C) for over 48hours, it was defined as a new FN event. We then collected, via chart review, all the FN events from the start of a patient's induction chemotherapy (or first chemotherapy at our institute) to the end of the final consolidation therapy.

## Definitions of the D-indices [4]

The indices investigated in this study are schematically illustrated in Fig 1. The D-index was calculated as the area between the neutrophil curve during grade 4 neutropenia and the line representing 500cells/μL. Accordingly, the D-index is noted to vary depending on the period of the calculation. The c-D-index was the cumulative D-index calculated from the start of grade 4 neutropenia to the onset of FN. If no FN occurred, the c-D-index was equal to the D-index. The c-D-index reflects an accumulation of neutropenia until FN (Fig 1) [4]. The total c-D-index was calculated using the period from the first chemotherapy to the onset of FN, whereas the total D-index was calculated using the period from the first chemotherapy to the final chemotherapy. These indices evaluated the additive effects of neutropenia, with the D-index providing a retrospective evaluation of the severity of neutropenia and the c-D-index offering a current evaluation of ongoing neutropenia. Further, the total D-index/total c-D-index is defined as the sum of the calculated overall D-indices (Fig 1) [4].

## Diagnostic procedure and protocol for infectious diseases

An infectious disease was diagnosed according to the following criteria: when the causative organism (bacterium or fungus) was identified from clinical specimens collected aseptically; when a pathologic organism (bacterium or fungus) that could cause clinical manifestations was isolated; or when an infectious disease was strongly suspected from the patient's clinical course and symptoms and was supported by certain serum markers of infection, such as β-D-glucan and procalcitonin. We then comprehensively categorized the patients' clinical symptoms into six focuses of infection according to clinical manifestations and culture results: oral cavity (stomatitis, toothache, and gingivitis), gastrointestinal (nausea, vomiting, diarrhea, and abdominal pain), respiratory tract (cough, sputum, dyspnea, and chest pain), skin (rash, swelling, flare, and painful skin, including in the device insertion area), bloodstream (blood culture positive but lacking in clinical manifestations), and unknown origin (blood culture negative and lacking in clinical manifestations or the coexistence of multiple clinical manifestations that would differentiate other diagnoses).

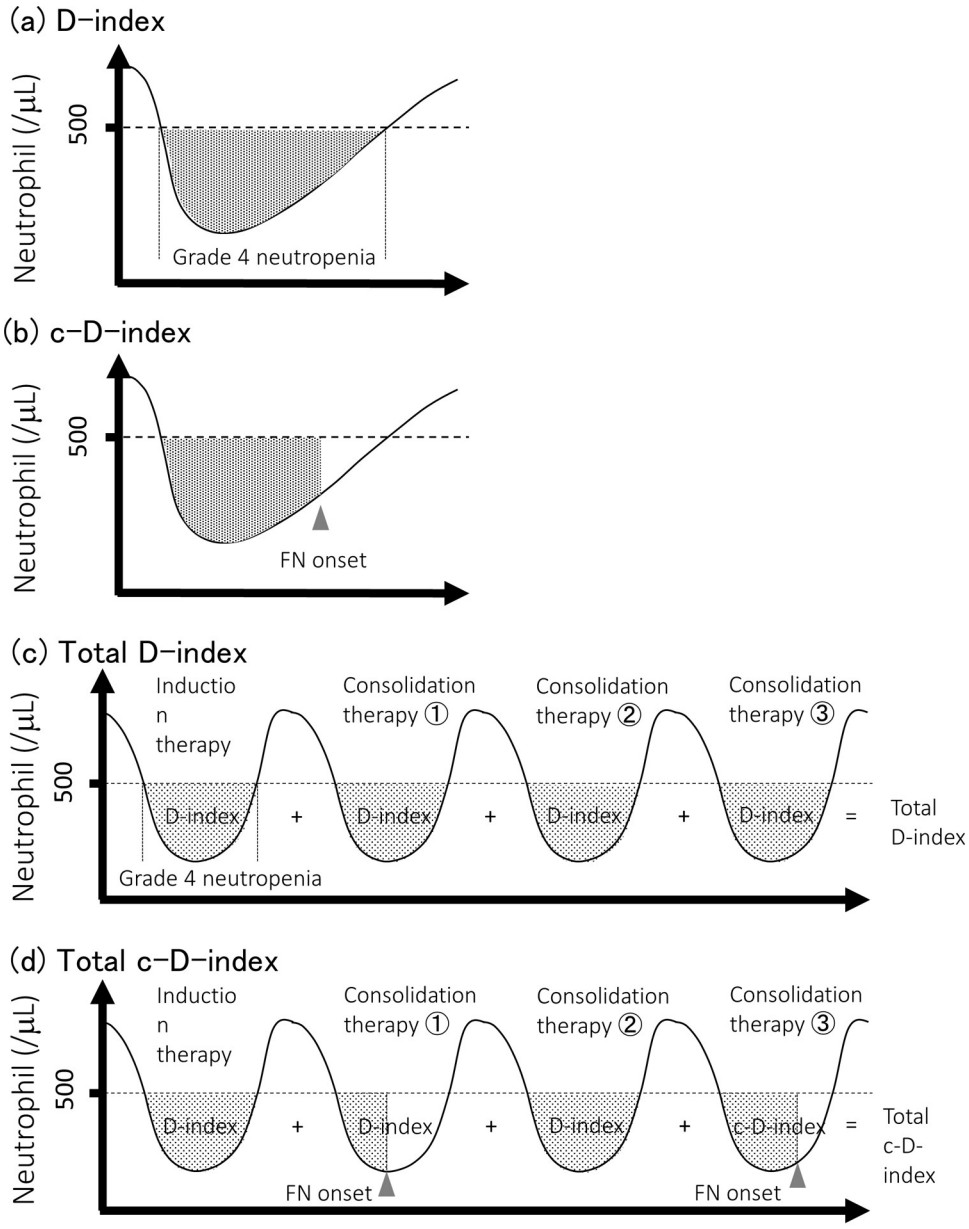

**Fig 1. Definitions of the four indices investigated.** The D-index, total D-index, cumulative D-index (c-D-index), and total c-D-index are depicted on the neutrophil count curve for a patient undergoing induction therapy and three periods of consolidation therapy. FN, febrile neutropenia.

## Statistical analysis

We used basic statistics to describe representative patient characteristics. A two-tailed paired Student's *t*-test (for parametric analysis) or a Mann–Whitney U test (for nonparametric analysis) was used to compare values between groups. A receiver operating characteristics (ROC) curve analysis was also performed to determine the cutoff value of the c-D-index at a given day after chemotherapy. We performed univariate analysis to establish explanatory factors for

clinical outcomes, including FN onset. Multivariate analysis was used to identify the risk factors contributing to the onset of FN, including the D-index and the c-D-index as potential factors. In multivariate analyses, we have evaluated patient background as an independent variable and onset of FN as a dependent variable using the regression model. P-values less than 0.05 were considered to indicate statistical significance. Patient background data included:

- Age (as a continuous variable)

- Sex (male or female)

- Chemotherapy regimen (anthracycline with cytarabine or high-dose cytarabine)

- Treatment phase (induction or consolidation)

- Disease risk (JALSG risk category)

- Disease status (complete remission or non-remission)

- Comorbidities (diabetes mellitus, grade 4 neutropenia at the start of chemotherapy, active infection, smoking history, past history of corticosteroid treatment, and prophylaxis for bacterial infection, each as an independent comorbid factor)

Statistical analyses were performed using SPSS version 19.0J software (SPSS Japan, Tokyo, Japan).

### Ethical issues

We collected the patients' information in chart review form with the approval of the Kagawa University Hospital Institutional Review Board (2020–110). We have also obtained patient consent to participate in our retrospective research by opt-out method. This study was conducted in accordance with the ethical standards of the responsible committee on human experimentation (Kagawa University Hospital Institutional Review Board) and of the Helsinki Declaration (1964, amended in 2008) of the World Medical Association. The patients' data were de-identified.

## Results

In total, 51 patients (25 women, 26 men; median age 53 years, range 18–74) met the eligibility criteria. These patients experienced 171 neutropenic events, of which 68 were during induction and 103 during consolidation chemotherapy, and 115 were FN and 56 afebrile episodes. The patients' accrual was shown in Fig 2 as a flow chart. The FAB classification was distributed as follows: M0, n = 2; M1, n = 6; M2, n = 21; M4, n = 4; M5, n = 6; M6, n = 6; and secondary AML, n = 6. The patients' characteristics are summarized in Table 1. The JALSG prognosis score indicated good risk in 16 subjects, intermediate risk in 19, poor risk in 14, and unknown in 2. The rate of response to induction chemotherapy was 56.8% (29/51), and 80.4% (41/51) of patients were alive at the end of follow-up. But one patient died of bacteremia due to *Stenotrophomonas maltophilia* during chemotherapy. Pathogens were isolated in 30 episodes (17.5%) among all 171 neutropenic events. The most common pathogens were *Streptococcus mitis* and *Klebsiella pneumoniae*, which were isolated in seven and three episodes, respectively. *Escherichia coli*, *Staphylococcus epidermidis*, and *Streptococcus intermedius* were detected in two cases each. The associations between c-D-index and infectious pathogens were compared according to the causative pathogen (gram-positive/negative organisms) or infection focus we detected. The comparison did not yield significant results (P = 0.285 and 0.443, respectively).

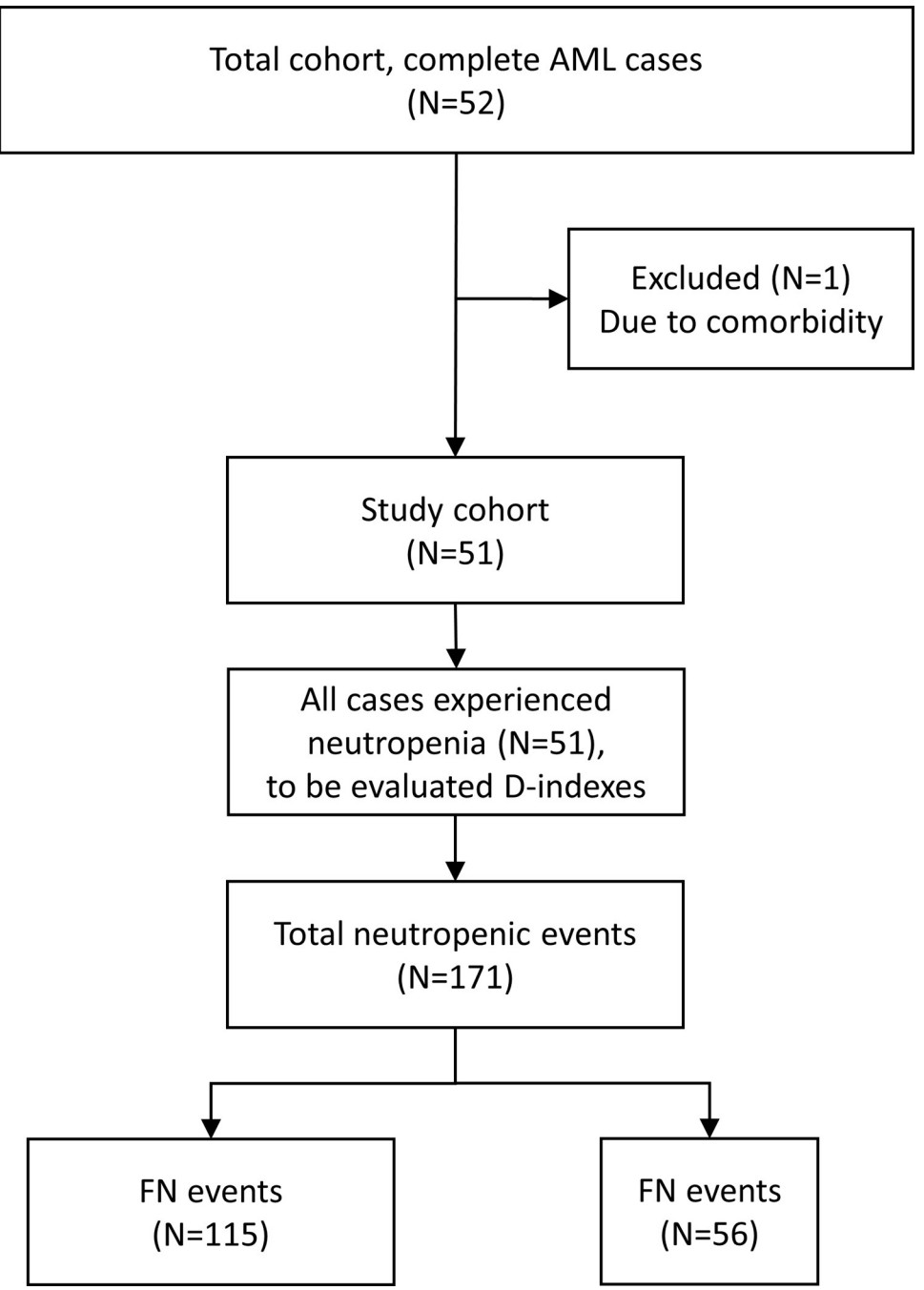

**Fig 2. Study patients flow chart.** We gathered 51 patients treated for acute myeloid leukemia at our institute. In total, 171 neutropenic events, of which 115 febrile neutropenia episodes were comorbid, were used in our study.

Next, we compared each D-index with and without FN (Fig 3). The D-index was found to be significantly higher in patients with FN (P < 0.01) (Fig 3A). The c-D-index on day 11 (c-D11-index) was significantly higher in patients with FN (P < 0.01) (Fig 3B). Nevertheless, the total D-index and the total c-D-index were not significantly different in patients with and without FN. ROC curve analysis defined a c-D11-index of 718 as the best cutoff value (Fig 4). The choice of day 11 was arbitrary, but we repeated the same analysis to find the best

**Table 1. Patient characteristics.**

| | | |
|---|---|---|
| Age (years) | Median | 53 |
| | Range | 18–74 |
| Sex | Male | 26 |
| | Female | 25 |
| JALSG risk | Good | 16 |
| | Intermediate | 19 |
| | Poor | 14 |
| | Unknown | 2 |
| FAB classification | M0 | 2 |
| | M1 | 6 |
| | M2 | 22 |
| | M4 | 4 |
| | M5a | 5 |
| | M5b | 0 |
| | M6a | 6 |
| | M7 | 0 |
| | Unknown | 6 |
| Diabetes mellitus | (+) | 8 |
| | (−) | 43 |
| Active infection | (+) | 7 |
| | (−) | 44 |
| Smoking history | (+) | 18 |
| | (−) | 28 |
| | Unknown | 5 |
| Past history of corticosteroid treatment | (+) | 0 |
| | (−) | 51 |
| Prophylaxis for bacterial infection | (+) | 46 |
| | (−) | 5 |
| Induction therapy | IDR + Ara-C | 36 |
| | DNR + Ara-C | 12 |
| | Unknown | 3 |
| Reinduction therapy | IDR + Ara-C | 9 |
| | DNR + Ara-C | 2 |
| | Other | 1 |
| | (−) | 39 |
| Consolidation therapy | High-dose Ara-C | 22 |
| | Anthracycline + Ara-C | 11 |
| | Unknown | 18 |

Abbreviations: JALSG, Japan Adult Leukemia Study Group; FAB, French–American–British classification; IDR, idarubicin; Ara-C, cytarabine; DNR, daunorubicin

sensitivity and specificity results, which were 68.5% and 68.4%, respectively, on ROC curve analysis (Fig 4). The cumulative incidence of FN was significantly higher in the group with c-D11-index ≥710 than in the group with c-D11-index <710 ($P < 0.01$) (Fig 5).

In the univariate analysis of risk factors for the onset of FN, c-D11-index and the presence of diabetes mellitus were identified ($P = 0.01$ for both factors) (Table 2). Multivariate analysis showed the same contributing factors, i.e., c-D11-index and the presence of diabetes mellitus, as risk factors for the onset of FN ($P = 0.01$ for both factors) (Table 2).

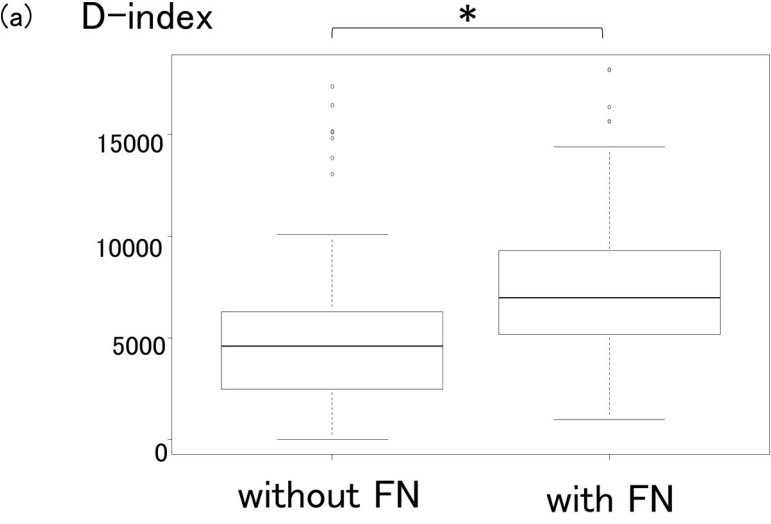

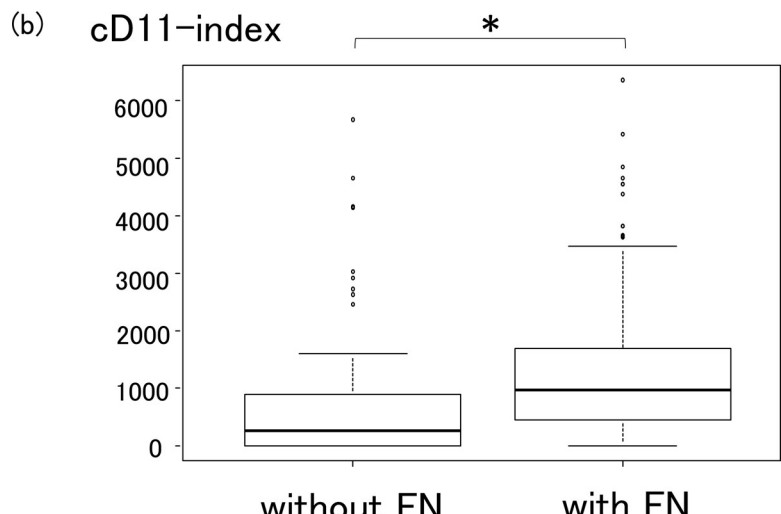

**Fig 3. Comparison of the D-index and the c-D-index in patients with and without febrile neutropenia (FN).** (a) The D-index was significantly higher in patients with FN (P < 0.001). (b) The c-D-index on day 11 (c-D11-index) was significantly lower in patients with FN (P < 0.001). *Statistically significant (P < 0.001).

## Discussion

In this study, we have investigated whether a set of various types of D-indices (D-index, c-D-index, total D-index, and total c-D-index) can predict FN. As per our findings, only the c-D11-index was significantly associated with FN onset. The D-indices are a group of evaluations of the accumulated deficit of the neutrophil count below 500 cells/µL over the neutropenic period. In this definition, the D-indices conceptually indicate "innate immunity deficiency." However, a prediction of FN itself using the D-indices as the clinical tool has not been evaluated. Our clinical question is how much the c-D-index changes the risk of FN. We then assessed the value of the c-D-index at 11 days after chemotherapy, specifically the cutoff value of ≥710, in predicting future onset of FN. This time and cutoff decision is deemed important because with it, we can take more intensive preventative action against the

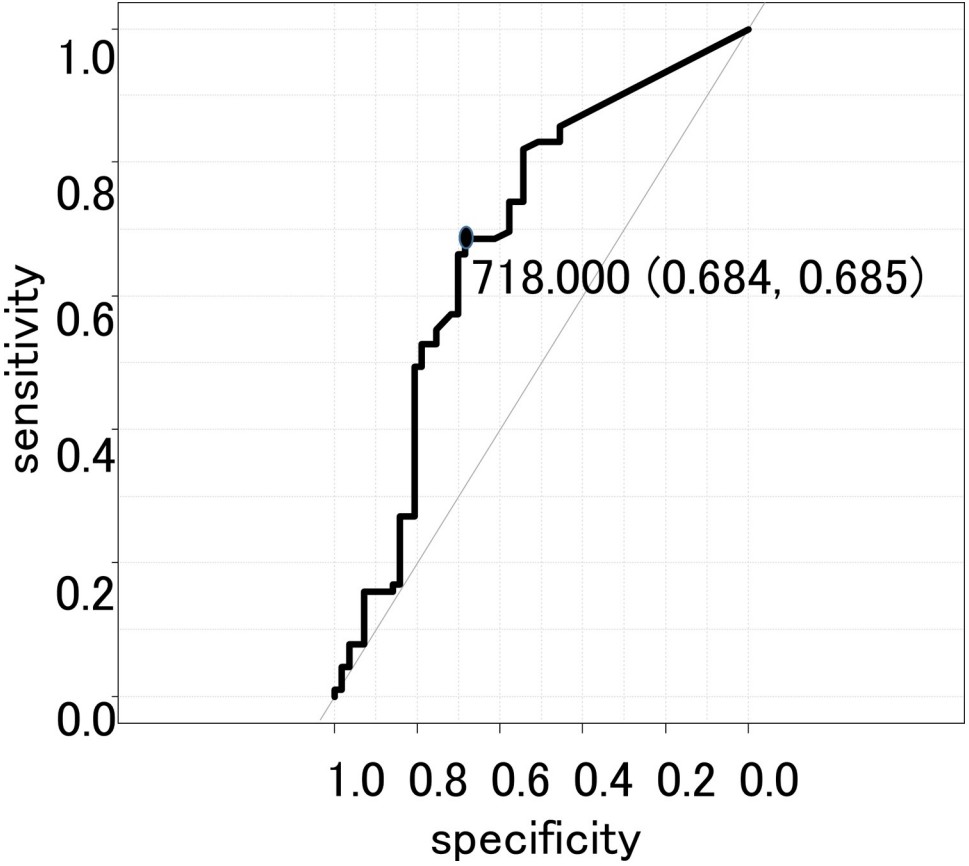

**Fig 4. Sensitivity and specificity validation analysis on receiver operating characteristics (ROC) curve.** ROC analysis defined a c-D11-index of 718 as the best cutoff value to discriminate between patients who develop FN after day 11 and those who do not. The sensitivity and specificity of ROC curve analysis were 68.5% and 68.4%, respectively.

development of opportunistic infection at the 11-day mark. Our results may expand the research into prospective interventional infection control according to the presence of FN risk or not.

In this consequent concept, the D-indices were investigated as potential predictors of infections resulting from neutropenia [7–11]. Reports have evaluated the relevance of the D-indices in the prediction of opportunistic infections. Indeed, the original report by Portugal et al. evaluated the efficacy of the c-D-index as a predictor of IFI [7]. They have reported that the c-D-index cutoff value of ≥5,500 is a sensitive predictor of IFI [7]. This original report was followed by further studies in this area [20, 21]. In both reports, a 9–10-day period of prolonged neutropenia was associated with a high risk of IFI. This finding reminds us that patients with antimicrobial-resistant FN after two or three rotations or additions of antibiotics would be at high risk of IFI. More recently, researchers have specified much more adaptation of the D-indices in IFI therapy [12–16]. In a large cohort study for assessing D-index utility in fungal treatment, proven invasive fungal infection was only seen in 1.01% (4/394) cases even after stem cell transplantation (SCT) [12]. They indicated the adequate cutoff value for c-D-index to predict IFI was 10,644, which is deemed too high. D-index-driven therapy exerted comparable [13] or noninferior [14] efficacy to empirical treatment among hematological malignancies, including SCT recipients. However, this predictive value was not shown in either the adult [15] or pediatric populations [16]. The result was controversial in evaluating the efficacy of the D-index in

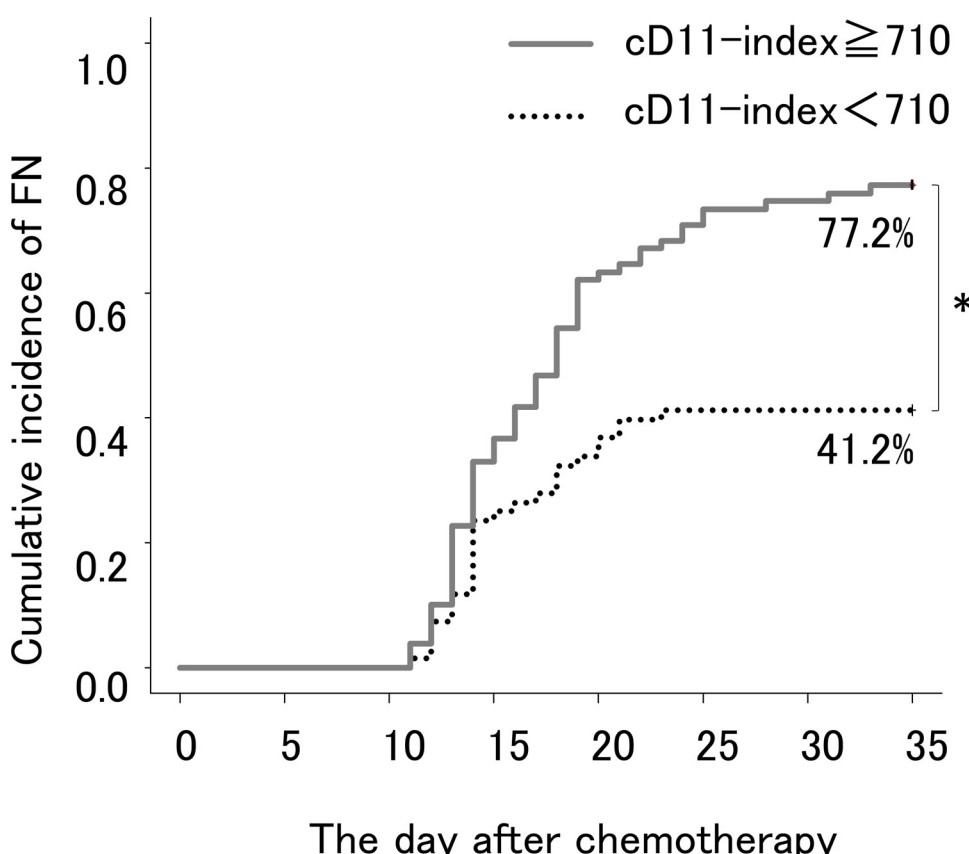

**Fig 5. Cumulative incidence of febrile neutropenia (FN) according to defined c-D11-index value.** The cumulative incidence of FN was significantly higher in the group with c-D11-index ≥710 (77.2%) than in the group with c-D11-index <710 (41.2%) (P < 0.001). *Statistically significant.

IFI treatment in hematological malignancy, including AML patients and SCT recipients. Thus, use of the D-index utility remains limited in IFI in neutropenic patients.

Another useful application of the D-indices is to estimate the site of location of the infection. FN has been associated with changes in—or the development of—local site- or organ-

**Table 2. Risk factors for the onset of febrile neutropenia.**

| Contributing factor | Patient characteristic | HR | P-value | HR | P-value |
|---|---|---|---|---|---|
| Patient characteristics and conditions | Age, ≥ median | 1.49 (0.98–2.27) | 0.06 | | |
| | Sex, female | 0.82 (0.54–1.25) | 0.35 | | |
| | Disease risk (JALSG), poor | 1.03 (0.62–1.72) | 0.90 | | |
| | Disease status, nonremission | 1.21 (0.79–1.85) | 0.39 | | |
| Chemotherapy regimen | High-dose Ara-C | 1.07 (0.70–1.64) | 0.75 | | |
| Past history | Diabetes mellitus | 2.07 (1.27–3.39) | 0.0038* | 1.94 (1.18–3.18) | 0.0087* |
| Clinical data | Absolute neutrophil count at the start of chemotherapy < 500/mL | 1.17 (0.68–2.02) | 0.56 | | |
| D-indexes | c-D11-index ≥ 710 | 2.41 (1.54–3.78) | 0.0001* | 2.34 (1.49–3.67) | 0.0002* |

Abbreviations: Ara-C, cytarabine; c-D-index, cumulative D-index.; HR, hazard ratio, JALSG; Japan Adult Leukemia Study Group.

Univariate analysis showed that diabetes mellitus and c-D11-index were contributing factors to the onset of febrile neutropenia. Multivariate analysis also found the same risk factors.

*Statistically significant.

specific infections, such as pneumonia, oral mucositis, gastrointestinal disease, bloodstream infection, or cutaneous infection. Regarding the infection focus, two later clinical investigations did not find the c-D-index to be a useful predictor of pulmonary infection during consolidation chemotherapy with high-dose cytarabine (Ara-C) [9] or after reduced-intensity SCT [10]. When we have assessed the predictive value of the D-indices for infection focus, they failed to be of significant value both in our previous reports [17] and this trial. One of the reasons for this was the small sample size. In order to advance in this field, it would thus be clinically helpful if c-D-index could predict more specific infectious diseases, not just IFI. However, infection in FN is poorly documented microbiologically [22, 23]. The D-indices might be indicators of specific kinds of infection.

One of the first series of reports regarding the D-index showed that pulmonary infections were predicted when the c-D-index was >5,500 in SCT recipients [8]. In contrast, a higher c-D-index did not predict bloodstream infection in that study [8]. Portugal et al. suggested that the c-D-index is not predictive of bloodstream infection because of this infection's early occurrence in neutropenia [24]. In our study, pathogens were definitively identified microbiologically in only 30 of 171 cases (17.5%). Prediction of infection in neutropenic patients with malignancies is clinically critical, because it directly contributes to effective antimicrobial therapy. Some advances have been made in the sensitive and accurate diagnosis of infections [23]. The development of these medical techniques and procedures will establish an easier approach to empirical and presumptive diagnosis-driven antimicrobial therapy [20, 25].

In other reports, oral and maxillofacial surgeons evaluated associations between the D-index and oral mucositis (or oral infection) and reported that a higher D-index was not associated with the development of odontogenic infection [26]. These results indicated that some specific infection sites may be associated with severe neutropenia. These infection sites might be influenced or affected by the regimens or doses used [16]. In our study, the onset of microbiologically proven infection was significantly associated with the use of high-dose Ara-C and the presence of diabetes mellitus. The onset of documented infection is thought to be confounded by the concurrent presence of diabetes mellitus, which is associated with poor hygiene in some individuals: poor oral hygiene, foot skin erosion, and so on. In fact, the prevalence of infection at such sites is noted to be higher among diabetic patients than among nondiabetic patients. This implies that diabetes is a universal risk factor for febrile episodes due to infection, including FN [27]; however, this has been found to be marginal or not significant in some trials [28]. The focus of the infection and the causative pathogen may be difficult to detect in FN patients.

In these situations, the D-index is an alternative approach to enable early treatment of underdiagnosed infection before the infection is proven or the patient becomes critically ill. Evidence is growing that the D-index is a useful and convenient tool for predicting IFIs [20]. Pediatric analysis [16] and regimen-related analysis [29] are also available from clinical trials. Recently, a similar evaluation that investigated the lymphocyte index (L-index) during chemotherapy reported that the D-index was more useful than the L-index for screening populations at risk for pulmonary infection [11].

The limitations of this study are that antibiotic data are lacking, it is a single-institute study, and the number of subjects is low. However, this is the first study to demonstrate prediction of FN by the c-D-index among AML patients during induction and consolidation chemotherapy. In conclusion, the c-D-index at day 11 can predict the onset of FN.

## Supporting information

**S1 Data.**
(XLSX)

## Author Contributions

**Conceptualization:** Hiroyuki Kubo, Osamu Imataki, Yukiko Hamasaki Kubo.

**Data curation:** Hiroyuki Kubo.

**Formal analysis:** Hiroyuki Kubo.

**Funding acquisition:** Osamu Imataki, Makiko Uemura.

**Investigation:** Hiroyuki Kubo, Osamu Imataki.

**Methodology:** Hiroyuki Kubo, Osamu Imataki, Makiko Uemura.

**Resources:** Hiroyuki Kubo, Osamu Imataki, Makiko Uemura.

**Supervision:** Osamu Imataki, Makiko Uemura.

**Validation:** Osamu Imataki, Makiko Uemura.

**Writing – original draft:** Hiroyuki Kubo, Osamu Imataki.

**Writing – review & editing:** Hiroyuki Kubo, Osamu Imataki.

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
