## [Decision Letter · Decision Letter 0]

15 Nov 2021

PONE-D-21-30375c-D-index can predict febrile neutropenia during chemotherapy in acute myeloid leukemiaPLOS ONE

Dear Dr. Imataki,

Thank you for submitting your manuscript to PLOS ONE. After careful consideration, we feel that it has merit but does not fully meet PLOS ONE’s publication criteria as it currently stands. Therefore, we invite you to submit a revised version of the manuscript that addresses the points raised during the review process.

We look forward to receiving your revised manuscript.

Kind regards,

Benigno C. Valdez, PhD

Academic Editor

PLOS ONE

Journal Requirements:

2. Please provide additional details regarding participant consent. In the Methods section, please ensure that you have specified (1) whether consent was informed and (2) what type you obtained (for instance, written or verbal). If your study included minors, state whether you obtained consent from parents or guardians. If the need for consent was waived by the ethics committee, please include this information.

 [The funders had no role in study design, data collection and analysis, decision to publish, or preparation of the manuscript.]

4. Thank you for submitting the above manuscript to PLOS ONE. During our internal evaluation of the manuscript, we found significant text overlap between your submission and the following previously published works, some of which you are an author.

https://link.springer.com/article/10.1007/s10147-018-01384-9

https://ashpublications.org/blood/article/134/Supplement_1/5190/425016/Lower-c-D-Index-in-Acute-Myeloid-Leukemia-with

Please revise the manuscript to rephrase the duplicated text, cite your sources, and provide details as to how the current manuscript advances on previous work. Please note that further consideration is dependent on the submission of a manuscript that addresses these concerns about the overlap in text with published work.

Reviewers' comments:

Reviewer's Responses to Questions

**Comments to the Author**

1. Is the manuscript technically sound, and do the data support the conclusions?

Reviewer #1: Yes

Reviewer #2: Partly

2. Has the statistical analysis been performed appropriately and rigorously? 

Reviewer #1: I Don't Know

Reviewer #2: No

3. Have the authors made all data underlying the findings in their manuscript fully available?

Reviewer #1: Yes

Reviewer #2: No

4. Is the manuscript presented in an intelligible fashion and written in standard English?

Reviewer #1: Yes

Reviewer #2: Yes

5. Review Comments to the Author

Reviewer #1: Very well-conducted research. However, some recommendations for improvement of the manuscript;

Comment 1: Introduction: Can it predict or differentiate the type of organism or severity of FN?

Comment 2: Method section:

a. Kindly mention the inclusion and exclusion criteria

b. Please add a flow chart of the patient inclusion

Comment 3: Result:

a. What was the association between the D-index and the duration of fever?

b. What were the differences in the D-index between different focuses of infection.

c. Was the index value compared with the type of organism or organ involved or severity of infection??

Comment 4: Discussion:

a. Did you find any correlation between this index and the site of infection?

b. Kindly focus the discussion on the findings of your study only.

Comment 5: Conclusion: what is the conclusion? significance of this information?

Comment 6: References:

a. Use more references from recent publications.

b. Consider adding some references from the developing countries where FN is rampant and the main cause of mortalities: e.g. Mishra K, Kumar S, Ninawe S, Bahl R, Meshram A, Singh K, et al. The clinical profile, management, and outcome of febrile neutropenia in acute myeloid leukemia from resource constraint settings. Ther Adv Infect Dis. 2021 Aug 4;8:20499361211036592. doi: 10.1177/20499361211036592.

Jandial A, Mishra K, Panda A, Lad D, Prakash G, Khadwal A, et al. Necrotising Ulcerative Gingivitis: A Rare Manifestation of Pseudomonas Infection. Indian J Hematol Blood Transfus. 2018 Jul;34(3):578-580. doi: 10.1007/s12288-018-0927-z.

Comment 7: Repetitions in the manuscript should be avoided.

Reviewer #2: c-D-index can predict febrile neutropenia during chemotherapy in acute myeloid leukemia

İntroduction:

- It is better to explain population as: ‘adult patients’.

- Definition of febrile neutropenia needs correction and reference.

- ‘We obtained informed consent to participate from all patients.’ .Due to retrospective nature of the study , it seems hard to take informed consent from all patients.

In results section:

- Was c-D-index different according to organism identified as the cause of febrile neutropenia?

- It is better to give values of c-D-index according to cultured gram positive / gram negative organisms.

‘We comprehensively categorized the patients’ clinical symptoms

into six focuses of infection according to clinical manifestations and culture results: oral

cavity (stomatitis, toothache, and gingivitis), gastrointestinal (nausea, vomiting, diarrhea, and

abdominal pain), respiratory tract (cough, sputum, dyspnea, and chest pain), skin (rash,

swelling, flare, and painful skin, including in the device insertion area), bloodstream (blood

culture positive but lacking in clinical manifestations), and unknown origin (blood culture

negative and lacking in clinical manifestations or the coexistence of multiple clinical

manifestations that would differentiate other diagnoses).’

- Was c-D-index different accordinf to focuses of infection?

- What about IFI? Was there any IFI cases in group?

In discussion section

-‘Portugal et al. reported a c-D-index cutoff value of ≥ 5500 for sensitive prediction of IFI.’Reference is needed.

-Discussion is inadequate.

6. PLOS authors have the option to publish the peer review history of their article (what does this mean?). If published, this will include your full peer review and any attached files.

Reviewer #1: **Yes: **Kundan Mishra

Reviewer #2: **Yes: **Kamile Arıkan

---

## [Author Response · Author response to Decision Letter 0]

31 Dec 2021

Response to Reviewers

To Reviewer 1:

1. Introduction: Can it predict or differentiate the type of organism or severity of FN?

Reply: Thank you for your comment. We, in our previous study, could not differentiate the type of organism and the focus of infection. And no other report predicts the severity of FN. Herein, the scope of our manuscript focused on predicting the onset of FN. We described this in the Introduction (Page 5, lines 4 to 12).

2. Method section:

a. Kindly mention the inclusion and exclusion criteria

b. Please add a flow chart of the patient inclusion

Reply: Thank you for your comment. (a) We have summarized the study inclusion and exclusion criteria in the Methods section (Page 6, lines 4 to 9). (b) Our study cohort is completely made up of consecutive AML patients treated at our institute. We recruited a total of 52 AML cases, but 1 case was not eligible for this study because of the patient’s comorbidity. Thus, we drew a simple flow chart for patient selection (new Figure 2).

3. Result:

a. What was the association between the D-index and the duration of fever?

b. What were the differences in the D-index between different focuses of infection.

c. Was the index value compared with the type of organism or organ involved or severity of infection??

Reply: Thank you for your comment. (a) The D-index is conceptually calculated as the sum of the products of neutropenic days and daily deficit of neutrophils below 500cells/�L. Therefore, the D-index is not directly associated with fever in its definition. However, prolonged neutropenia can lead to a higher D-index. Clinically, it is anticipated that a higher D-index would be associated with prolonged fever, especially neutropenic fever. In our study cohort, among 171 neutropenic events, we observed 115 FN and 56 afebrile episodes. We added this information in the Results section (Page 11, lines 4 to 5). (b) We evaluated the difference in the D-index by infection focus; however, no significant association was noted between D-index and infection focus (Page 11, lines 17 to 18). (c) Similarly, we could not find any significant relationship between the D-indices and the type of pathogen or infection focus (Page 11, lines 15 to 18).

4. Discussion:

a. Did you find any correlation between this index and the site of infection?

b. Kindly focus the discussion on the findings of your study only.

Reply: Thank you for your comment. (a) We evaluated the difference in the D-index between infection loci; however, there was no significant association between D-index and infection focus (Page 11, lines 15 to 18). (b) We extensively revised our Discussion section.

5. Conclusion: what is the conclusion? significance of this information?

Reply: Thank you for your comment. Our conclusion is simple: The c-D-index at day 11 can predict the onset of FN (Page 17, line 5).

6. References:

a. Use more references from recent publications.

b. Consider adding some references from the developing countries where FN is rampant and the main cause of mortalities: e.g. Mishra K, Kumar S, Ninawe S, Bahl R, Meshram A, Singh K, et al. The clinical profile, management, and outcome of febrile neutropenia in acute myeloid leukemia from resource constraint settings. Ther Adv Infect Dis. 2021 Aug 4;8:20499361211036592. doi: 10.1177/20499361211036592.

Jandial A, Mishra K, Panda A, Lad D, Prakash G, Khadwal A, et al. Necrotising Ulcerative Gingivitis: A Rare Manifestation of Pseudomonas Infection. Indian J Hematol Blood Transfus. 2018 Jul;34(3):578-580. doi: 10.1007/s12288-018-0927-z.

Reply: We added ten recent publications, including the article you suggested above (new reference numbers #6). However, the case report which you suggested (Jandial et al. Indian J Hematol Blood Transfus 2018) was not suitable for our manuscript. Thank you for your kindness.

7. Repetitions in the manuscript should be avoided.

Reply: Thank you for the relevant remark. We checked and deleted the redundant description in our manuscript.

 

To Reviewer 2:

1. İntroduction:

a. It is better to explain population as: ‘adult patients’.

b. Definition of febrile neutropenia needs correction and reference.

c. ‘We obtained informed consent to participate from all patients.’ Due to retrospective nature of the study, it seems hard to take informed consent from all patients.

Reply: Thank you for your comment. (a) We have specified the study population as adult patients (Page 5, lines 8 and 12). (b) We defined the febrile neutropenia with the reference (Page 7, lines 9 to 12). (c) Our study was approved by the Institutional Review Board at our institute, and the consent was obtained via opt-out method in posting the study rationale. We revised the description of patients’ consent in the Ethical Issues section (Page 10, lines 11 to 12).

2. Results section:

a. Was c-D-index different according to organism identified as the cause of febrile neutropenia?

b. It is better to give values of c-D-index according to cultured gram positive / gram negative organisms.

‘We comprehensively categorized the patients’ clinical symptoms into six focuses of infection according to clinical manifestations and culture results: oral cavity (stomatitis, toothache, and gingivitis), gastrointestinal (nausea, vomiting, diarrhea, and abdominal pain), respiratory tract (cough, sputum, dyspnea, and chest pain), skin (rash, swelling, flare, and painful skin, including in the device insertion area), bloodstream (blood culture positive but lacking in clinical manifestations), and unknown origin (blood culture negative and lacking in clinical manifestations or the coexistence of multiple clinical manifestations that would differentiate other diagnoses).’

c. Was c-D-index different according to focuses of infection?

d. What about IFI? Was there any IFI cases in group?

Reply: Thank you for your comment. (a)(b) Among the patients with FN, documented infections were identified only in 30 episodes out of 171 FN events. Then, the comparison according to the infection sites and the causative pathogens (gram-positive/negative organisms), which is not designated to the statistical comparability, did not prove significance (P=0.285 and 0.443, respectively). We described this result in the Result section (Page 11, lines 15 to 18). (c) We evaluated the difference in the D-index based on infection focus. However, no significant association was noted between D-index and infection focus. (d) Basically, in FN patients, causative pathogens are unknown throughout the treatment. In our cohort patients, the proven invasive fungal infection cases were quite rare (only a single case).

3. discussion section

a. ‘Portugal et al. reported a c-D-index cutoff value of ≥ 5500 for sensitive prediction of IFI.’ Reference is needed.

b. Discussion is inadequate.

Reply: Thank you for this relevant remark. (a) We quoted the manuscript published by Portugal (Page 13, line 19). (b) We extensively revised our Discussion section.

---

## [Decision Letter · Decision Letter 1]

24 Jan 2022

c-D-index at day 11 can predict febrile neutropenia during chemotherapy in acute myeloid leukemia

PONE-D-21-30375R1

Dear Dr. Imataki,

We’re pleased to inform you that your manuscript has been judged scientifically suitable for publication and will be formally accepted for publication once it meets all outstanding technical requirements.

Kind regards,

Benigno C. Valdez, PhD

Academic Editor

PLOS ONE

Additional Editor Comments (optional):

Reviewers' comments:

Reviewer's Responses to Questions

**Comments to the Author**

1. If the authors have adequately addressed your comments raised in a previous round of review and you feel that this manuscript is now acceptable for publication, you may indicate that here to bypass the “Comments to the Author” section, enter your conflict of interest statement in the “Confidential to Editor” section, and submit your "Accept" recommendation.

Reviewer #1: All comments have been addressed

2. Is the manuscript technically sound, and do the data support the conclusions?

Reviewer #1: Yes

3. Has the statistical analysis been performed appropriately and rigorously? 

Reviewer #1: I Don't Know

4. Have the authors made all data underlying the findings in their manuscript fully available?

Reviewer #1: Yes

5. Is the manuscript presented in an intelligible fashion and written in standard English?

Reviewer #1: Yes

6. Review Comments to the Author

Reviewer #1: The manuscript has been nicely revised by the authors. Heartiest congratulations to all the authors.

7. PLOS authors have the option to publish the peer review history of their article (what does this mean?). If published, this will include your full peer review and any attached files.

Reviewer #1: **Yes: **Kundan Mishra

---

## [Editor Report · Acceptance letter]

4 Mar 2022

PONE-D-21-30375R1 

c-D-index at day 11 can predict febrile neutropenia during chemotherapy in acute myeloid leukemia 

Dear Dr. Imataki:

I'm pleased to inform you that your manuscript has been deemed suitable for publication in PLOS ONE. Congratulations! Your manuscript is now with our production department. 

Kind regards, 

on behalf of

Dr. Benigno C. Valdez 

Academic Editor

PLOS ONE